# *Drosophila* Corazonin Neurons as a Hub for Regulating Growth, Stress Responses, Ethanol-Related Behaviors, Copulation Persistence and Sexually Dimorphic Reward Pathways

**DOI:** 10.3390/jdb9030026

**Published:** 2021-07-05

**Authors:** Ziam Khan, Maya Tondravi, Ryan Oliver, Fernando J. Vonhoff

**Affiliations:** 1Department of Biological Sciences, University of Maryland, Baltimore County, Baltimore, MD 21250, USA; zkhan3@umbc.edu (Z.K.); mayat2@umbc.edu (M.T.); ryan.oliver@mail.huji.ac.il (R.O.); 2Department of Ecology, Evolution, and Behavior, The Hebrew University of Jerusalem, Jerusalem 9190401, Israel

**Keywords:** neuropeptide, growth, dimorphism, addiction, alcoholism, eating disorder, insect disease model

## Abstract

The neuronal mechanisms by which complex behaviors are coordinated and timed often involve neuropeptidergic regulation of stress and reward pathways. Recent studies of the neuropeptide Corazonin (Crz), a homolog of the mammalian Gonadotrophin Releasing Hormone (GnRH), have suggested its crucial role in the regulation of growth, internal states and behavioral decision making. We focus this review on Crz neurons with the goal to (1) highlight the diverse roles of Crz neuron function, including mechanisms that may be independent of the Crz peptide, (2) emphasize current gaps in knowledge about Crz neuron functions, and (3) propose exciting ideas of novel research directions involving the use of Crz neurons. We describe the different developmental fates of distinct subsets of Crz neurons, including recent findings elucidating the molecular regulation of apoptosis. Crz regulates systemic growth, food intake, stress responses and homeostasis by interacting with the short Neuropeptide F (sNPF) and the steroid hormone ecdysone. Additionally, activation of Crz neurons is shown to be pleasurable by interacting with the Neuropeptide F (NPF) and regulates reward processes such as ejaculation and ethanol-related behaviors in a sexually dimorphic manner. Crz neurons are proposed to be a motivational switch regulating copulation duration using a CaMKII-dependent mechanism described as the first neuronal interval timer lasting longer than a few seconds. Lastly, we propose ideas to use Crz neuron-induced ejaculation to study the effects of fictive mating and sex addiction in flies, as well as to elucidate dimorphic molecular mechanisms underlying reward behaviors and feeding disorders.

## 1. Introduction

Corazonin (Crz) is an 11 amino acid neuropeptide present in most arthropods, named after its ability to accelerate heartbeat in cockroaches [1]. It forms a peptide family together with the mammalian Gonadotropin Releasing Hormone (GnRH) and its insect orthologue, the adipokinetic hormone (AKH), based on their shared sequence features and the relatedness of their receptors [2,3,4,5]. Numerous studies have indicated that Crz regulates development under starvation conditions by interacting with stress response and homeostatic pathways. This is consistent with additional roles of Crz neurons and Crz-dependent signaling pathways regulating food intake, reward, and ethanol-related behaviors. This review summarizes recent advances in our understanding of Crz neurons and their roles in development, metabolism, and reward behavior.

Furthermore, the neuronal mechanisms behind *Drosophila* mating remain an exciting area of investigation as it is a complex behavior that requires animals to coordinate between courtship and copulation. Males first attempt to court a female and, if successful, will initiate copulation by mounting the female and engaging the genitals. Copulation usually culminates with male ejaculation. An interesting phenomenon observed in males is the high persistence at the start of copulation despite life-threatening stimuli. While it is known that male copulatory persistence decreases over time, the mechanisms regulating this motivational process remain incompletely understood. In recent literature, four male-specific Crz interneurons within the abdominal ganglia (Abg) of the ventral nerve cord (VNC) are emerging as key regulators for male mating behavior. Activation of Crz neurons using opto- and thermogenetic tools induces ejaculation in male flies [6,7,8]. In particular, precise molecular interactions within Crz neurons have been elegantly described to regulate copulation duration, representing the first neuronal mechanism for timing intervals longer than a few seconds [9]. Understanding of Crz-dependent molecular and neuronal networks might help elucidate the hedonic values of specific components of mating and other addictive behaviors. While mating and ethanol consumption are known to activate the Neuropeptide F (NPF) reward pathway, the anatomical and functional connectivity between reward, mating and ethanol-related pathways remain to be elucidated. Consistently, an exciting recent study has demonstrated that activation of Crz-neurons is pleasurable to male flies and increases *npf* transcript levels [8], suggesting a potential role of Crz neurons as key players in the regulation of reward-behavior and addiction-like behaviors.

Lastly, we discuss ideas for possible novel research directions based on the known functions of Crz neurons, such as investigating the impact of the use of “fictive mating” on an animal’s nervous system and behavior. Although the impact of Crz neuron research is still at its initial stages, we propose novel research directions that may potentially have clinical relevance. For example, we discuss the possibility of using Crz neurons as a sexually dimorphic model to further investigate the molecular mechanisms underlying addictive behavior and feeding disorders, based on known interactions with reward pathways and ethanol-dependent behaviors as well as interactions with the insulin signaling pathways and food intake, respectively. Such ideas are supported by the high levels of conservation between the insect and the human neuroendocrine systems [10,11,12,13], as well as by the observation that about 75% of genes associated with human diseases are thought to have a functional homolog in the fly [14].

## 2. Developmental Processes Involving Crz-Neuron Function Are Required for the Regulation of Diverse Physiological Mechanisms

### 2.1. Corazonin Regulates Systemic Growth, Feeding Behavior, Stress, and Homeostasis

In *Drosophila*, Crz expression has been detected during late embryogenesis, restricted to three different neuronal types: dorso-lateral (DL), dorso-medial (DM), and Crz neurons in the ventral nerve cord (vCrz) (Figure 1). The last two groups undergo apoptosis during metamorphosis, while DL neurons survive into adulthood [15]. Crz is proposed to act on short range Crz receptor (CrzR) neurons located in the brain and ventral cord [8,16,17,18]. In addition, Crz is thought to exert hormone-like peripheral effects on distantly located cells, e.g.on CrzR cells in the fat body, whose activation depends on Crz release into the hemolymph [3,16,19,20]. Interestingly, Crz neurons represent a clear example of sexual dimorphism at the cellular and molecular level, as a group of four neurons develops *de novo* during early pupal development in the abdominal ganglion (abgCrz) of male flies. Such abgCrz neurons are involved in male-specific behavior (described below) and are absent in female flies [15].

Multiple studies indicate that Crz plays a functional role in development and survival mechanisms, especially during stress and starvation conditions. In the adult brain, Crz expression is restricted to the endocrine dorso-lateral peptidergic (DLP) group, which consists of a cluster of six to eight neurons per brain lobe [15]. DL-Crz neurons, co-expressing the short neuropeptide F (sNPF), stimulate insulin producing cells (IPCs) in the pars intercerebralis and are known to affect carbohydrate and lipid levels in response to starvation [20]. Similarly, larvae under nutrient restriction conditions show increased Crz and sNPF levels in DL neurons to promote larval development into puparation, a process requiring signaling via ER-Ca^2+^ stores [22]. Additional lines of evidence support a role of Crz in sensing nutrients, including the observations that adult DL-Crz neurons express the fructose receptor Gr43a [23], as well as their involvement in larval sugar sensing [24]. Moreover, interactions between Crz signaling and neuropeptides known to influence water and ionic homeostasis have been reported. DL-Crz neurons express receptors for neuropeptides diuretic hormone 31 (DH31), DH44, and AstA [19,25]. Additionally, Crz inhibits a specific group of CrzR neurons that express the neuropeptide CAPA, with the purpose to suppress diuresis and ensure osmotic homeostasis [17]. Therefore, the role of Crz in nutritional stress is widely supported [3,19,26].

In addition to its involvement in survival conditions, several studies indicate a role of Crz in the regulation of feeding and growth. Downregulation of Crz, CrzR, and sNPF decreases food intake [26,27], which regulates growth in flies by interacting with insulin/IGF signaling and *Drosophila* insulin-like peptides (DILPs). Genetic ablation of IPCs and loss of function of the *Drosophila* insulin/IGF1 receptor, DInr, causes developmental delay and growth retardation [28,29]. By contrast, overexpression of DInr and Dilp2 increases cell size [29], and chronic activation of Crz neurons [30] as well as increased sNPF levels [27] promote food intake and lead to heavier flies. Interestingly, extended survival and decreased aging, similar to the phenotype caused by caloric restriction [31,32], has been observed following decreased Crz signaling via cell ablation or silencing of Crz neurons, sNPF knockdown, or *Dlnr* mutations [20,33,34]. sNPF acts via its sNPF receptor, sNPFR1, expressed in olfactory receptor neurons to enhance olfactory sensitivity and attractiveness to food odors such as vinegar in starved flies and to promote foraging [35]. Furthermore, an elegant study described feeding as a competitive behavior of escape (or fleeing) by examining the emotional primitives of valence and persistence based on *Drosophila* state-dependent behavioral responses [36]. Therefore, Crz may serve an important role in the decision-making process between feeding and fleeing, although direct evidence for the latter remains to be obtained.

### 2.2. Systemic Growth Inhibition by DL-Crz Modulation of Ecdysone

Systemic growth is regulated by 20-hydroxyecdysone during development, hereby referred to as ecdysone, which is the active form of a steroid hormone secreted by the prothoracic gland and has been implicated in a myriad of powerful and diverse signaling pathways [37,38]. Ecdysone’s modulation by DL-Crz neurons related to systemic growth is discussed in this section.

Basal levels of ecdysone have been shown to negatively control system body growth by acting as an antagonist to general insulin signaling via interaction with the fat body, a structure homologous to the liver in vertebrates [39]. This relationship was further validated when silencing of Ecdysone receptors (EcR) resulted in increased pupal size [39]. Prothoracicotropic hormone (PTTH) has long been implicated in the regulation of peak levels of ecdysone production at the prothoracic gland [40], and, more recently, in low levels as well via Warts (Wts)/Yorkie (Yki)/microRNA *bantam* (*ban*) signaling [41,42,43]. Anatomical and functional analyses demonstrate that DL-Crz cells activate CrzR expressed in PTTH neurons [17]. Both silencing of Crz neurons and CrzR knockdown in PTTH neurons resulted in increased pupal size. Interestingly, this effect was phenocopied by knockdown of the octopamine receptor *Oamb* in Crz neurons, suggesting that octopaminergic neurons act upstream of Crz neurons in the regulation of systemic growth [17].

The antagonistic effects of the Crz-PTTH-ecdysone pathway to insulin driven systemic growth have clear implications in obesity research. In humans, numerous studies have found that birth weight is strongly correlated with obesity later in life [44]. Furthermore, it has been found that adult obesity is positively correlated with the presence of famine in the first two trimesters of pregnancy [45], but negatively correlated with famine occurring in the last trimester [46,47]. As this axis is active in development, it has the potential to provide insight to unravel the molecular mechanisms underlying obesity present in early childhood before they become classically exhibited in adulthood.

Moreover, a recent connectomics study focused on mapping synaptic connections within the neuroendocrine system in first instar larvae [13]. They found that CO_2_ readings from the trachea of the animal are integrated by tracheal dendritic neurons (TD) identified by the same study. The information then passes through interneurons T1–T3 to DM and DL Crz-expressing neurons. However, further analysis shows that Crz neurons are mostly innervated not by T1–T3, but rather by the interneuron Munin 2 which provides over 50% of the input to Crz neurons. Munin 2 interneurons convey information from pharyngeal sensory neurons, which sense food intake [13]. Crz neurons were found to mostly innervate the corpora cardiaca, but also the corpus allatum and the prothoracic gland to a lesser degree [13]. The pharyngeal input is consistent with the role of Crz neurons in modulating systemic growth, while the input from T1–T3 interneurons with CO_2_ information is consistent with the many roles of Crz regarding stress described elsewhere in the paper.

### 2.3. Ecdysone-Driven Apoptosis of vCrz Neurons during Metamorphosis

As discussed above, basal levels of ecdysone mediate systemic growth through insulin signaling. However, ecdysone has another role in development—surges of ecdysone at the beginning of pupariation mediate maturation through apoptosis signaling, among other pathways.

A hallmark of neural development in species ranging from mice [48] to humans [49], is the overbuilding of synaptic connections and the subsequent elimination of superfluous synapses, neurites, and entire neurons [50,51]. Therefore, this combination of refinement processes at the synaptic (via synaptic pruning mechanisms) and the neuronal level (via programmed cell death, PCD) is one of the key players during the development of precise connectivity within neuronal networks [52]. This refinement process contributes to the increased priority placed on plasticity during development [53] and on efficiency in the streamlined adult brain. Eight pairs of vCrz neurons are eliminated during *Drosophila* metamorphosis [54]. Specifically, six to seven hours after the pupa is formed, vCrz neurons undergo PCD, a process that is inhibited by the ablation of ecdysone signaling [54].

PCD in vCrz neurons occurs via two mechanisms [55] (Figure 2). In both, ecdysone binds to either the EcR-B1 or EcR-B2 isoform of the ecdysone receptor-B (EcR-B) following a surge in the steroid hormone at the beginning of pupariation [54]. In the first pathway, this binding occurs redundantly in astrocyte-like or cortex glial cells [56] causing the secretion of the ligand Myoglianin (Myo), which binds to the Babo-A receptor in vCrz neurons, activating TGF-β signaling and leading to apoptosis [55]. Loss of function of Myo suppresses vCrz PCD, phenocopying the effects of EcR blockade by the expression of an EcR dominant negative [55]. In the second pathway, ecdysone binds to EcR in the vCrz neuron itself, which forms a complex with the nuclear receptor, Ultraspiracle (Usp) [57]. Downstream of EcR-Usp, *grim* acts as the main death gene, whereas *sickle* and *reaper* play only a minor role in vCrz PCD [55,58]. Interestingly, no suppression of vCrz PCD is observed following Usp knockdown, suggesting that Usp function is not required by the EcR-dependent *myo* expression for vCrz PCD [54]. The formation of EcR-Usp after ecdysone binding to the EcR-B1 isoform transforms the astrocyte-like glia into a phagocyte that clears the corpses of vCrz neurites and other cellular debris produced by vCrz PCD during metamorphosis [55]. By contrast, vCrz cell bodies are disposed of via a different pathway independent of ecdysone [55,59].

Endocrine-disrupting chemicals (EDCs) are substances that bind to steroid hormone receptors, thereby inhibiting steroid hormones, such as ecdysone [60], which could interfere with the pruning process resulting in developmental impairment. In fact, because these chemicals are able to cross the blood–brain barrier, relatively low-dose exposure to EDCs is already thought to cause impairments in neural development and have been linked to the development of several neurological disorders [61]. Future studies could use vCrz neurons to examine the effects of these chemicals on apoptosis and synaptic pruning. DL neurons represent a natural control for these experiments because they persist into adulthood whereas vCrz neurons are fated for destruction [54]. The ecdysone-driven differential phagocytosis of vCrz neurites and cell bodies could be an invaluable tool to study the mechanisms underlying astrocyte-dependent precise phagocytosis during development as well as their role in the targeted phagocytosis of other cells.

## 3. Role of Crz Neurons in Ethanol-Related Behaviors

### 3.1. Developmental Transcription Regulation and Neuronal Activity in Crz Neurons Modulate Sensitivity to Ethanol Sedation

During development, proper regulation of transcriptional programs in Crz neurons are known to modulate ethanol-induced behaviors. Ethanol-induced sedation can be quantified by measuring the temporal dynamics of a loss of righting reflex (LORR—the number of flies unable to right themselves) when flies are exposed to high levels of ethanol vapor [62]. The role of Crz neurons in ethanol sedation in *Drosophila* was first identified in a genetic screen that tested different GAL4 drivers for their potential to alter sensitivity to ethanol-induced sedation by expressing an RNAi construct to downregulate the gene apontic (*apt*) [7]. *apt* encodes a Myb/SANT-containing transcription factor and its role in ethanol sensitivity was identified by a previous genetic screen in the same study that tested mutant fly lines with different P-element insertions for their potential to affect ethanol sedation sensitivity [7]. Loss of function mutations in *apt* as well as *apt* knockdown in Crz neurons reduced ethanol sensitivity. Interestingly, *apt* function in Crz neurons was required during development as well as in adulthood to regulate sedation sensitivity [7]. Reduced ethanol sensitivity was also observed following *crz* knockdown as well as by cell ablation and silencing of Crz neurons. Consistently, thermogenetic activation of Crz neurons increased ethanol sensitivity [7]. These results suggest that a small group of neurons is able to regulate complex behaviors, involving specific signaling pathways that are in the process of being elucidated.

### 3.2. Ethanol Recovery and Consumption Are Regulated by Crz Neuron Function

Activity of Crz neurons has also been linked to ethanol-related behaviors such as recovery from ethanol-induced sedation (“hangover phenotype”) and ethanol consumption. A more severe hangover phenotype was observed following cell ablation of Crz neurons as well as in animals bearing loss of function mutations in the *crzR* gene as compared to controls [16]. A mechanism for ethanol metabolism and tolerance is shown to involve the Crz-dependent regulation of alcohol dehydrogenase (ADH) and aldehyde dehydrogenase (ALDH). Whereas cell ablation of Crz neurons decreased only ALDH activity, *CrzR* mutant flies showed decreased ADH and ALDH activity. These results suggest two CrzR-dependent signaling pathways: a Crz-dependent upregulation of ALDH activity as well as a Crz-independent downregulation of *Adh* transcription. For the latter, PKA function in CrzR neurons was shown to regulate *Adh* expression, with no effect on *Aldh* transcript levels [16].

By contrast, the role of Crz neurons in ethanol consumption is thought to involve mechanisms related to mating and reward (discussed in Section 4). The enhanced preference to consume ethanol-supplemented food in sexually deprived males [63] was suppressed by optogenetic activation of Crz neurons [8]. One proposed mechanism suggests that Crz neuron activation is rewarding to male flies, increases transcription of the neuropeptide F (*npf*) gene (described below), and reduces ethanol consumption [8]. Interestingly, the suppression of ethanol consumption in sexually deprived males is accomplished by only the activation of the four abdominal ganglionic male abgCrz, whereas activation of only brain Crz neurons had no effect. Therefore, the Crz-dependent regulation of ethanol consumption is proposed to involve different mechanisms than the ones underlying ethanol sensitivity [8].

Neuropeptide F is the fly homolog of the mammalian Neuropeptide Y and is not closely related to sNPF. NPF and sNPF are involved in distinct signaling systems, activate different receptors, and play non-redundant roles [64]. In fact, whereas colocalization of Crz and sNPF has been observed in adult brains as described above [20], no overlap was observed between brain NPF and Crz neurons, indicating mutually exclusive populations of neurons [8]. Similar to NPY in mammals, NPF levels represent different motivational states in *Drosophila*. Rewarding events such as mating and ethanol intoxication increase *npf* levels [8,63] whereas exposure of females to parasitic wasps [65,66] and sexual deprivation in males decreases NPF signaling [8,63]. In addition, reduced NPF/NPFR1 signaling in adults decreases sensitivity to alcohol sedation, whereas overexpression of NPF causes the opposite effect [67]. Consistently, a role of NPF/NPY signaling in ethanol tolerance has also been observed in *C. elegans* and mice [68,69]. Thus, Crz and NPF are thought to be involved in similar molecular pathways, a relationship that is further supported by the observation of Crz-dependent regulation of *npf* transcription levels (discussed below). However, at the transcriptomic level, differences between Crz and NPFR1 neurons have been observed in regard to their RNA editing profiles [70]. Intriguingly, a recent study demonstrated that reduced RNA editing via knockdown of ADAR (adenosine deaminase acting on RNA) in NPFR neurons enhanced male–male social interactions, whereas ADAR knockdown in Crz neurons had moderate effects [71].

## 4. Crz Neuron Activity Modulates Mating-Related Features Such as Ejaculation, Copulation Duration, and Reward-Behaviors

### 4.1. Crz Functions as a Hub Regulating Ejaculation within the VNC

Studies have previously implicated the VNC in copulation [72,73]. In particular, the abdominal ganglion (Abg) is heavily involved in copulation [74]. Many of these neurons express the sex determining factor fruitless (Fru) [75]. Fru is known to regulate various aspects of male mating behavior including courtship. Activation of the four male-specific, *fru*-expressing abgCrz neurons is sufficient to induce ejaculation [6,7,8]. Injection of synthetic Crz peptide also induces ejaculation making Crz the first ejaculation inducing peptide identified in insects [6]. The proejaculatory function of Crz seems to be conserved among insects [76]. Manipulations of Crz Receptor projection neurons (CrzR) have similar effects as those in Crz neurons [6]. Silencing of CrzR neurons prevents ejaculation while activation induces it, suggesting that Crz neurons promote ejaculation through downstream CrzR neurons. Consistently, projections of CrzR neurons have been described to innervate male reproductive organs [6].

Whereas some upstream targets of brain Crz neurons have been identified, presynaptic partners of abgCrz neurons remain less well studied. As described above, brain Crz neurons are known to express specific diuretic neuropeptide receptors [19,25], supporting their role in homeostasis and stress response. By contrast, little is known about connections with abgCrz neurons and their regulation of ejaculation and copulatory behavior. Although a recent study described the sensory pathways and the presynaptic interneurons of neurosecretory cells in the larval brain [13], we currently do not know how sensory information is relayed to Crz neurons in the adult VNC and thus, sources of presynaptic input into Crz neurons are of particular interest for future studies. One potential candidate is the cholinergic mechanosensory neurons of the genitalia [74]. The sensilla of these neurons are located on the male genital claspers and function early in copulation to promote proper genital coupling. Another option is the dsx/vGlut motor neurons that control the phallic and perphallic organs and function to promote genital coupling [74]. During copulation, the mechanosensory neurons feedback to the dsx/vGlut neurons to promote correct posture. Future studies could focus on identifying presynaptic inputs into abgCrz neurons to elucidate how Crz neurons are able to integrate various types of information and regulate copulation-related phenomena.

### 4.2. Different Degrees of Conservation in Mechanisms and Molecules Regulating Ejaculation That Relate to Crz-Dependent Signaling

Several lines of evidence support a role of serotonin in ejaculation. A small population of male-specific Abg CrzR neurons are serotonergic and innervate the male reproductive organs [6]. Consistently, injection of 5-HT induced ejaculation, whereas silencing of either CrzR-GAL4 neurons or serotonergic neurons inhibited sperm transfer and led to infertility [6]. In humans, serotonin is important for patients with premature ejaculation [77,78]. Selective Serotonin Reuptake Inhibitors are becoming drugs of interest for the treatment of premature ejaculation [79]. Additionally, premature ejaculation is associated with polymorphisms in the Tryptophan Hydroxylase 2 gene [80], which plays a fundamental role in brain serotonin synthesis [81]. Premature ejaculation is also linked to the serotonin-transporter-linked polymorphic region 5-HTTLPR, which is a degenerate repeat polymorphic region in SLC6A4, the serotonin transporter gene [82,83]. Consistently, 5-HTT knockout mice have altered ejaculation [84,85].

Many of the factors that activate ejaculation in flies have also been implicated in humans. This suggests that the ejaculatory neural circuitry found in flies may be evolutionarily conserved [86]. Although a direct proejaculatory role for the Crz-related GnRH in mammals has not been identified, intriguing similarities between Crz and GnRH functions have been observed. GnRH is a key component of the hypothalamic–pituitary–gonadal (HPG) axis in mammals and its role in puberty and hypogonadism have been explored. GnRH is also linked with sexual behavior in both male and female mammals, by directly regulating testicular and pituitary function (reviewed in [87]). Effects of GnRH manipulation on ejaculation have been reported by several studies, which have not necessarily focused on a direct relation with ejaculation. Rats treated with a GnRH antagonist impaired ejaculation and fertility [88], administration of GnRH reduced ejaculation latency in gonadally intact male rats [89], and rats missing the s-GnRH neurons show a loss of ejaculation [90]. Interestingly, NPY injection in male rats increased *Gnrh1* transcript levels, modulating sexual behavior [91]. These results underscore the urgent need to further characterize the distinct levels of conservation between Crz and GnRH at the functional, molecular, and behavioral level.

### 4.3. Crz-Independent Function of abgCrz Neurons

Although Crz peptide is sufficient to induce ejaculation, its necessity remains questionable. Whereas silencing and ablation of abgCrz neurons results in a complete loss of male fertility [6], neither *crzR*-null [16] nor *crz* loss-of-function mutants have notable fertility defects [17,92]. This suggests that abgCrz neurons are essential for male fertility but Crz peptide may be dispensable for this function. Crz-peptide independent functions of abgCrz neurons are further supported by the observation that not all manipulations of CrzR neurons phenocopy similar manipulations of Crz neurons [6]. This could result from an intrinsic function of abgCrz neurons or from signaling to additional downstream targets through the expression of additional neuropeptides in abgCrz neurons. Determining whether abgCrz neurons are capable of inducing ejaculation in *crz*- and *crzR*-null backgrounds would establish if Crz peptide is necessary for ejaculation. Furthermore, this would also enable the identification of additional functions for abgCrz neurons in the copulation circuitry that may be independent of the Crz peptide. Besides ejaculation, there are several additional discrepancies between Crz and CrzR neuron manipulations. For example, triacylglyceride (TAG) levels were affected after *crz* knockdown, while *crzR* knockdown in the periphery had no effect, suggesting that Crz-dependent regulation of lipid metabolism may involve other signaling systems [26]. Additionally, whereas *CrzR* mutant flies showed altered ADH and ALDH activity (and a stronger hangover phenotype), ablation of Crz neurons only affected ALDH activity [15]. Lastly, Crz neurons regulate copulation duration via mechanisms independent of downstream CrzR neurons (discussed below).

Support for non-CrzR signaling of abgCrz neurons comes from the observation that abgCrz neurons express acetylcholine (ACh) [6]. Larval Crz neurons located in the brain may also utilize ACh in addition to several other neurotransmitters [93]. Transcriptomic analysis of the larval brain revealed Crz neurons express genes needed for the packaging and synthesis of ACh, glutamate, and GABA. Additionally, these neurons showed expression of vesicular monoamine transporter which is needed for vesicular packaging of monoamines such as serotonin, dopamine, and octopamine. The expression of ACh is particularly interesting as ACh has long been implicated in male fertility and ejaculation in a wide range of species. In rats, 66% of animals treated with muscarine ejaculated [94]. In *C. elegans*, ACh is released from sensory motor-neurons, which facilitates movement of sperm from the seminal vesicles to the vas deferens [95]. Furthermore, cholinesterase inhibitors have also been shown to induce ejaculation in men with spinal cord injuries [96,97]. More recently, Botulinum-A toxin has been proposed as a treatment for premature ejaculation [79]. Whereas ACh-dependent induction of ejaculation in flies remains directly untested, there is evidence that the cholinergic circuitry is crucial for male fertility. Genetic feminization of the cholinergic neurons causes a drastic decrease in fertility and results in ablation of ~12 male-specific cholinergic neurons in the Abg [86]. It is possible that the four abgCrz neurons were among them, but Crz-specific feminization has yet to be performed. It would be worth identifying how many of the male-specific cholinergic neurons are essential for ejaculation.

### 4.4. Crz Function as a Timer to Regulate Copulation Duration and Animal Persistence

Persistence in animals can be easily described as a continued behavior despite either high risk or low reward. Persistent behavior in *C. elegans* is regulated by the neuropeptide Pigment-Dispersing Factor (PDF) and serotonin [98]. In mice, activation of 5-HT neurons in the dorsal raphe nucleus promoted increased persistence [99]. The use of a selective 5-HT2AC ligand was also found to promote enhanced persistence in mice [100]. Hunger drives persistence in flies [101] and activation of male-specific P1 interneurons promotes a state of persistent courtship behavior or aggression [102]. Interestingly, although persistence and copulation duration are intrinsically linked, both processes are described to be regulated by distinct sets of neurons and different molecular pathways. The decreased persistence throughout copulation observed in males is known to be regulated by eight sexually dimorphic GABAergic neurons and other dopaminergic neurons in the VNC [103]. By contrast, while copulation duration is set by the sexually dimorphic serotonergic Fru neurons in the abdominal ganglion called sAbg-1 [75], silencing of Crz neurons increases copulation duration [6]. The increase in copulation duration observed by the genetic feminization of cholinergic neurons could also be explained by this result [86]. Interestingly, even though activation of CrzR neurons *in copulo* is sufficient to cause ejaculation and shortens copulation duration, activation of Crz neurons only induces ejaculation without altering copulation duration [6]. These findings establish that copulation duration is not determined by ejaculation as one may initially predict. Furthermore, these findings show that copulation duration is regulated by Crz neurons independent of downstream CrzR neurons [6].

A mechanism by which Crz neurons are able to intrinsically regulate copulation duration was recently identified [9]. Expression of a constitutively active Ca^2+^/calmodulin-dependent protein kinase II (CaMKII-CA) in Crz neurons is sufficient to extend mating duration. CaMKII-CA expression does not alter ejaculatory ability when Crz neurons are activated suggesting that CaMKII specifically regulates copulation duration and not ejaculation. Copulation duration is dependent on CaMKII signaling in Crz neurons where it acts as a molecular clock regulating male persistence. CaMKII signaling in Crz neurons regulates a 6 min time window to coordinate ejaculation and persistence thus making it the first neuronal mechanism for timing intervals longer than a few seconds. Activation of Crz neurons during copulation reduced persistence, indicated by the male’s immediate response to noxious threats. Thus, Crz neurons do not determine copulation duration *per se*, they instead function as a motivational ‘switch’. Once activated, the animals are switched out of a state of perpetually high motivation to a state in which persistence diminishes over time. Such neuronal mechanisms are of broad interest in terms of understanding how animals are driven to engage in one behavior over another [9].

### 4.5. Activation of Crz Neurons Is Pleasurable

The notion that sex in animals is rewarding is supported by numerous studies [104,105], including the fact that both male and female rats experience an orgasm-like response (reviewed by [106]). However, an intriguing question involves the hedonic value of specific components of mating. The multifaceted roles associated with Crz neurons make them a strong candidate to promote different aspects of reward [8]. In fact, when flies can choose between an unilluminated zone or a zone that optogenetically stimulates Crz neurons, males show a strong preference for the stimulation zone, suggesting that activation of Crz neurons, and therefore ejaculation, has a positive valence. A similar effect is observed by the activation of CrzR neurons. Furthermore, after conditioning training involving optogenetic activation of Crz neurons in the presence of an odor, trained animals show a preference for the associated odor even in the absence of light, suggesting that Crz neuron activation is sufficient to induce appetitive learning. Interestingly, optogenetic activation of NPF neurons phenocopied the flies’ preference for the illuminated side of the chamber [107]. Consistently, an interaction between Crz neuron activity and the NPF-reward pathway was determined. *npf* levels are similar in virgin and in Crz-silenced males, whereas mated and Crz-activated males show increased *npf* transcript levels [8]. Furthermore, Crz stimulation was sufficient to increase *npf* levels, which is likely sufficient to promote a postcopulatory molecular state without physical copulation with a female [8]. This idea is further supported by the fact that Crz neuron activation reduces ethanol consumption, a trait that involves decreased NPF levels and is characteristic of males that successfully mate [63].

## 5. Future Directions

### Possible Use of Fictive Mating to Study Sex Addiction and Feeding Disorders in Drosophila

The effects of copulation on animals remains an active field of study. Male Crz neurons offer a highly tractable tool with which to model excessive sexual activation. The possibility to induce ejaculation using opto- and thermogenetic tools [6,8] might provide a similar rewarding experience to the stimulated male fly even in the absence of physical contact with a female. We defined such an experimental scenario as “fictive mating”. Fictive mating following prolonged activation of Crz neurons enables a level of stimulation that would normally be prevented because of satiety. Mated male flies are known to have reduced lifespans and altered oxidase activity [108]. Therefore, this tool could be used to measure ongoing costs of reproduction [109]. PPLab (protocerebral posteriolateral dopaminergic cluster neuron 2ab) neurons represent a candidate to be likely affected by fictive mating as they have been shown to be affected by aging and involved in age-related decline in courtship [110]. An additional interesting experiment would be to test whether prolonged activation of Crz neurons in males leads to altered connections in their brain reward centers, since such flies would be consistently ejaculating in the absence of available mates and lacking visual and mechanosensory stimulation. Furthermore, we also hypothesize that such a fictive mating scenario will significantly alter male behavior and social interactions. It is possible that fictive mating leads to hypersexualized males, which may be less likely to interact with females. Evidence has shown that males that have previously mated have reduced pre-copulatory success [111]. While these experiments will not answer the question of whether ‘sex addiction’ is a disease, it will help us to understand how artificial sexually activity can alter an animal’s brain state. The use of a tool like Flyception2, which allows brain activity imaging in copulating male flies, may be particularly useful in investigating such behavior [112].

The existence of sex addiction is a controversial topic [113,114]. The Diagnostic and Statistical Manual of Mental Disorders (DSM) previously had a classification, but the most recent edition does not include a description of such a disorder. One of the major reasons non-substance related behaviors are no longer considered addictions is that there is no clear evidence for tolerance or withdrawal. Prevalence of such a disorder is hard to predict but around 80% of those who have such a condition are male. In mammals, activity of within the nucleus accumbens (NAc), a region known for its connection with addiction, was found to be a predictor of appetitive behaviors such as food and sex [115], and hippocampal input to the NAc shell modulates its activity to properly regulate goal-directed behaviors involving plasticity-dependent mechanisms [116]. Based on their described roles in food intake, ethanol sensitivity and copulation, Crz neurons represent a good cellular candidate to unravel the molecular mechanisms underlying reward-behavior and addiction.

Activation of the small population of the serotonergic Abg CrzR neurons as well as injection of 5-HT induced ejaculation [6]. Interestingly, activation of serotonergic neurons promoted aggressive behavior in male flies [117]. Additionally, as mentioned above, Crz neurons express the octopamine receptor Oamb [16]. Octopamine, the insect homolog of norepinephrine, is known to promote aggression by activating Oamb receptors in the male-specific aSP2 [118]. Although a direct role of Crz neurons in *Drosophila* aggression has not been directly observed [119], Crz injection in ants promoted hunting behavior, increased cricket biting, and slightly (but not significantly) increased the number of aggressive individuals [120]. Multiple studies proposed octopamine as the signal for reward in insects [121,122,123,124]. Such dual roles in aggression and reward seem to be conserved as aggression can activate reward centers in the mammalian brain, supporting the long-discussed concept of aggression reward and addiction [125,126], but evidence to support the connection between aggression, reward and Crz in flies is still lacking.

*Drosophila* has a long history as a model for studying addiction [127,128] and its application in the study of anxiety and other behaviors is just emerging [129,130]. Since a link between anxiety and eating disorders is well documented in developing children [131,132], one additional idea for possible future research involves the use of Crz neurons to study the molecular basis and developmental processes of feeding disorders [133,134]. Examples might include a molecular characterization of genes identified in human GWAS [135], including the Insulin Receptor Substrate 1 (IRS1) gene that showed sexual dimorphism in allelic effects at fasting insulin [136]. Eating disorders are thought to be more prevalent in women, but it remains unclear whether they are under-reported in general by a gender or whether sex-differences play a role within molecular pathways. As mentioned above, feeding in flies is regulated by the brain Crz neurons present in females and males, which is unlikely to be affected by the male-specific abgCrz neurons. However, manipulations of Crz neurons showed significant dimorphic effects in stress and metabolic responses including differences in triglyceride levels [33]. Therefore, future research could test whether Crz and its interacting signaling pathways (insulin, sNPF, NPF; [137]) would regulate feeding behavior in flies in a sexually dimorphic manner [138,139], with the potential to expand our understanding of fundamental mechanisms that might underlie eating disorders.

## 6. Conclusions

As described above, Crz neurons may serve as unique models to investigate the biological basis of addictive and reward behaviors based on their involvement in food intake, stress response, homeostasis, mating, and ethanol-induced behaviors. Curiously, Crz function has been associated with insect behaviors [140] that might resemble all four basic and most primal drives (the four f’s): feeding, fleeing, fighting, and mating [141,142] though its role in *Drosophila* fighting and aggression remains elusive. The implication of male Crz neurons as a hub for regulating particular components of male-specific behaviors including ejaculation, copulation duration, and NPF-dependent reward, makes them an attractive model to study sexually dimorphic neuronal network function and modulation of behavior. The high degree of interactions and similarities shared between Crz and other conserved molecules (such as GnHR) and peptidergic (such as NPF and sNPF) signaling pathways underlying goal-directed behaviors and addiction, strongly suggest that research on Crz neurons may expand our understanding of conserved mechanisms in the mammalian brain, and thus, be relevant to human health.

## Figures and Tables

**Figure 1 jdb-09-00026-f001:**
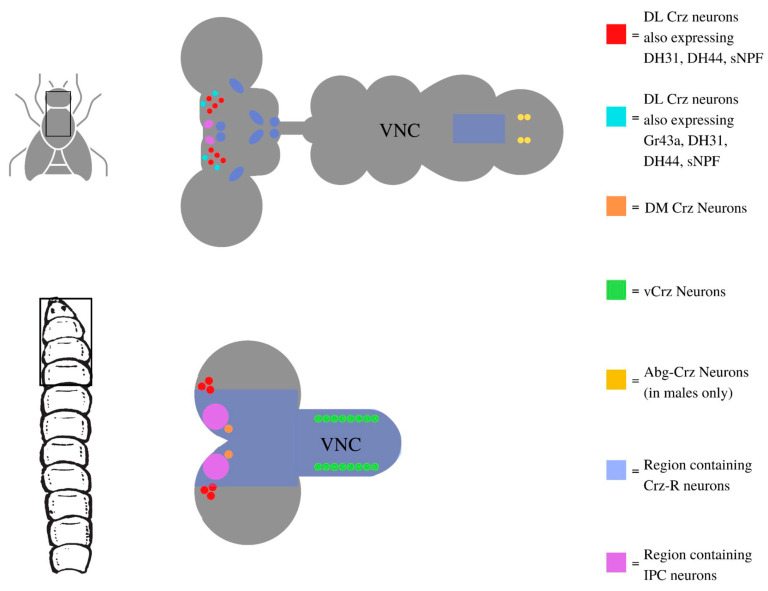
Diagram showing the populations of Corazonin (Crz) expressing neurons in the nervous systems of third-instar larvae and adult flies. In third-instar larvae, one dorsomedial (DM) neuron and three dorsolateral (DL) neurons are present in either hemisphere of the brain, and eight pairs of neurons (vCrz) are present on either side of the ventral nerve cord. DH31-R1 and DH44-R1 expression is present in all of these cells, while sNPF is only present in the DM and DL neurons, but not the vCrz neurons [21]. In adults, 6–8 dorso-latero-posterior (DLP) neurons are present on either hemisphere of the brain, and 4–6 Abg-Crz neurons are present in the abdominal ganglion in males only. DH31-R1, sNPF, and DH44-R1 expression is present in the DLP cells but not the AbgCrz neurons. Cells that also express Gr43a are indicated in cyan, however it should be noted that the number and locations of these cells vary. General regions in the adult and larval nervous system that contain Crz receptors (CrzR) are indicated in blue, while regions containing IPC neurons are indicated in magenta. The specific locations of additional neurons in the fly neurosecretory system (such as sNFP or IPCs) has been recently reviewed [11]. Relative locations of larval CrzR neurons are based on [15,16]. It is worth mentioning that different observations have been reported for CrzR expression in the adult brain. Whereas Zer-Krispil et al. (2018) [8] reported CrzR expression in numerous fat cells that surround the brain, a recent study by Zandawala et al. (2021) [17] showed expression of CrzR in brain neurons.

**Figure 2 jdb-09-00026-f002:**
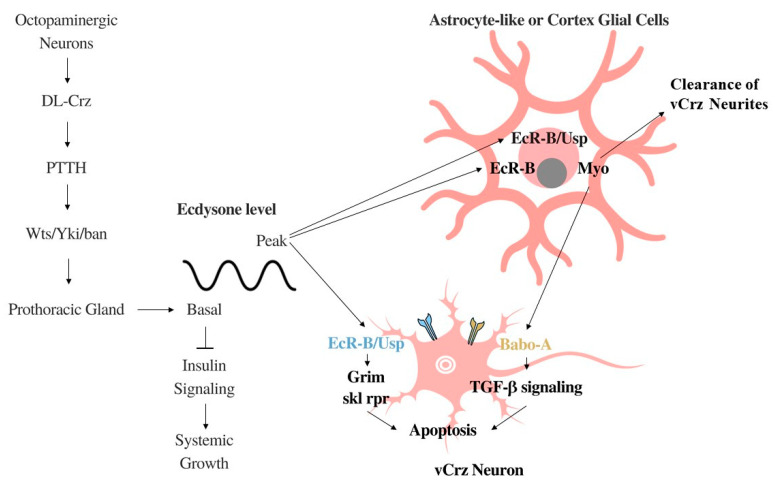
DL-Crz regulation of basal ecdysone production as well as the two pathways leading to vCrz apoptosis and one regarding the clearance of the resulting neural debris. Beginning at the left of the figure, DL-Crz neurons express octopamine receptors and innervate PTTH cells to modulate basal ecdysone production in the prothoracic gland. In the middle and right portion of the figure, basal ecdysone acts on the fat body to inhibit insulin signaling, thereby negatively controlling systemic growth. When ecdysone levels peak, it binds to either Astro-like or cortex glial cells at the EcR-B receptor to trigger the production of Myo, which binds to the Babo-A receptor in vCrz neurons causing apoptosis via TGF-β signaling. Ecdysone also binds to the EcR-B/Usp receptor complex which transforms astrocyte-like cells only into phagocytes which clear neural debris produced by Crz neuron apoptosis. Finally, peak ecdysone can also act on vCrz neurons directly via the EcR-B/Usp receptor complex to cause apoptosis through the major and the two minor death genes, *grim* and *sickle* (*skl*)/reaper (*rpr*), respectively [55,58].

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
