# Peer review of "Drosophila Corazonin Neurons as a Hub for Regulating Growth, Stress Responses, Ethanol-Related Behaviors, Copulation Persistence and Sexually Dimorphic Reward Pathways"

_jdb, 2021, doi:10.3390/jdb9030026_

Round 1
Reviewer 1 Report
Khan et al summarizes a recent progress of the study about Corazonin (Crz) neurons in Drosophila. Crz is a conserved GnRH-family hormone among various animal species and has pleiotropic roles in development, reproduction and physiology. Therefore, it is worth explaining the latest molecular mechanism understood in fly research for a broader range of readers. In addition, the authors discuss how much Crz knowledge is applicable for our medical sciences. This reviewer would appreciate this effort for spreading the value of fly basic research to all other animal scientists. To make this manuscript better understood for more readers, especially for non-Drosophila researchers, this reviewer suggests that the authors modify the manuscript as follows;
(1) The schematic drawings of the anatomical positions of Crz neurons would be helpful for understanding the section 2.1. In the larval brain, three types of Crz neurons in the dorso-lateral region, the dorso-medial region, and the ventral nerve cord, should be clarified. In the adult brain, abCrz neurons of male flies should be illustrated.
(2) The section 2.2 describes apoptosis of Crz neurons. This reviewer is afraid if this section does not fit the flow of story, because the role of Crz was not related to the apoptosis signaling pathway. Thus, the title 2 "Roles of Crz in development, apoptosis" may not be appropriate. As the section 2.1 describes growth and feeding, and the section 2.3 is about the regulatory mechanism of growth, the reviewer suggests the section 2.3 follows the section 2.1.
(3) On the page 6, line 144, "ecdosyne in vertebrates" may be a typo.
(4) On the page 6, line 148, the reference "Okamoto and Yamanaka, 2020" may be wrong. This paper describes the functional analysis of ecdysone importer, but not vCrz neurons. This reviewer strongly asks the authors to make sure if all the references are placed precisely once again.
(5) On the page 6, line 160-161, microRNA bantam should be italic. The reference needs to include Boulan et al., 2013, Current Biology 23: 473-478, instead of "Columbani et al., 2005" (Colombani et al., 2005).
(6) On the page 6, line 167, the authors need to explain more why the Crz-PTTH-ecdysone pathway has implications in diabete and obesity research.
(7) On the page 8 and 9, line 241, This reviewer cannot see any logical connection between Fruitless, GABAergic neurons and abCrz neurons in ejaculation. The term "ejaculation" appeared suddenly in the context. When some readers may think copulation includes ejaculation in a general sense, it is confusing. To avoid such a confusion, the authors need to explain the definition of terms "copulation", "courtship", "sex", "mating" and "ejaculation". Indeed, this reviewer cannot understand what is "fictive mating".
(8) Some abbreviations need to be clarified somewhere in the text, including PDF (Line 342), sAbg-1 neurons (Line 348), and PPLab neurons (Line 402).
(9) Recently a paper published in eLife (Huckesfeld et al., 2021;10:e65745) reported that Crz neurons are involved in carbon dioxide responsive network. This reviewer recommends the authors to describe this paper in this manuscript.
Reviewer 2 Report
RE: jdb-1253146
The review by Khan et al., is a comprehensive summary corazonin neuropeptide in Drosophila, focusing on its role in development, apoptosis, food intake, stress response, ethanol and reward-related behaviors, ejaculation and copulation timing, and offer future directions. The review is up to date, interesting and offers important contribution to the field.
Comments:
- The review is submitted for publication in Journal of Dev. Biol and consists of a portion that explains the development of CRZ neurons, but the majority of the text is focused on adult behavior. In addition, the focus of the review is not clear, it spans from neuro-developemnt, roles in larva, adult, dimorphism, reward, ethanol and addition and the flow and logics between the different sections is not clear.
- The title is too complex and not accurate- only a small portion of CRZ neurons are dimorphic and they are not an actual part of the reward system. I suggest changing the title.
- The introduction is not written in a clear manner, the connection between the paragraphs and the logic is not clear.
- section 2: A diagram of CRZ circuitry (brain+VNC) will be helpful for the readers. I suggest adding the markers mentioned (Dh31, sNPF, IPCs, Gr43a) on the relevant neurons.
- The existence of CRZ-receptor neurons in the brain is not clear. Please include their relative location in the scheme.
- Lines 108-112 are not relevant to CRZ, I suggest removing.
- Regarding developmental apoptosis and pruning. The authors use both terms together as if they are one process. Please explain the role of pruning and the role of apoptosis in each of the neuro-developmental process associated with CRZ neurons (see libe 123-126). In addition, the manuscript by Oren-Suissa et al is less relevant since it describes the use of pruning for the formation of dimorphic circuits, this is not the case for CRZ neurons. I suggest referring to papers from the Schuldiner group.
- Line 171, the connection to the previous section is not clear, are we still dealing with development related issues? This section needs to be connected better and also the ethanol related behaviors are not very clear.
- Line 173 and 190 - missing ref.
- Line 234- I an not sure that this paragraph is required since you already described the dimorphic CRZ neurons, what is the added value of introducing Fru, and GABA neurons?
- Line 245- Consider mention that Crz-R neurons innervate the males reproductive system.
- The rational of the paragraph starting at line 250 is not clear.
- Lines 379-381- This part describes that flies prefer to reside in zone that triggers optogenetic activation of crz neurons. This is also true to the activation of Crz-R neurons. Furthermore, in Shao et al 2017 (Dissection of the Drosophila neuropeptide F circuit using a high-throughput two-choice assay) flies in which NPF neurons are activated, display preference for the illuminated side. Consider mention this to strengthen the connection between Crz and NPF.
- line 439, the connection between aggression, reward and CRZ is not supported by evidence.
- Line 444, the connection between dimorphic CRZ neurons and feeding is not relevant. As you mention feeding is mediated via brain CRZ neurons.
- Section 3.2 is over-reaching, it is beyond CRZ and shifts the review towards regulation of ejaculation. Is this the focus of the review. Overall, there are too many ideas, and the synthesis between them is not working well.
Round 2
Reviewer 2 Report
The revised version is improved in many respects. Still I have couple of points needed to be addressed:
- The review mentions types and names of CRZ and CRZ receptor neurons and their co-expression with other markers at the brain and VNC of adult flies. The provided illustration does not contain all the details. In addition, looking on the reference mentioned, there are no CRZ-R neurons within the brain.
- Line 527- CRZ neurons are not part of the reward system, they induce ejaculation.
- Line 496, the reference of Heberlein is not relevant.
